# Solving In-Table Prediction Problems by Deep Neural Networks with Performance Evaluation Using Synthetic Data Journal Submissions

## Abstract

Tabular deep learning (TDL) leverages neural networks (NN) to extract patterns from tabular data. Traditional TDL methods follow a supervised learning paradigm, where a target feature is explicitly given. In this work, however, we explore a different approach by employing deep NNs to learn relationships among individual columns within a given table. We investigate whether NNs can predict the values of arbitrarily selected columns in a given table based on the remaining known columns. We call this problem In-Table Prediction (ITB), which is slightly different from table imputation methods and the pretraining task of TDL. Three potential usage scenarios are identified, which, to our best knowledge, have not been extensively studied in the literature. A self-supervised learning approach is applied to address this problem by randomly selecting columns to be masked out and used as learning targets. This work focuses on tabular datasets containing only continuous features. To handle missing values in continuous features, a novel neural layer is proposed to embed both numerical and empty values. Synthetic data is generated based on predefined column relationships, with empty values inserted using two distinct mechanisms. Additionally, an adapted masking strategy is employed to create test data. Performances of three NN architectures, namely MLP, Resnet and Transformer, are evaluated using the generated synthetic data. We conclude that, the attention-based structure outperforms the other two networks, when a sufficiently large number of training examples is available and a relatively large embedding length is chosen.

## 1 Motivation and contribution

Since the breakthrough of deep learning technology on image classification problems (cf. Krizhevsky et al. (2012) and references therein), deep learning (DL) has gained tremendous successes in advancing the methodologies in Computer Vision (CV) and Natural Language Processing (NLP). To extend this success to tabular data, a lot of researches have been conducted recently (Somvanshi et al., 2024; Borisov et al., 2024; Gorishniy et al., 2021; Rubachev et al., 2022). In comparison to CV and NLP, where image data and language data are homogeneous, tabular data are heterogeneous and structured. A table usually contains multiple columns, which may have different data types, and multiple rows, which are organized in a structured way. Although the semantic meaning of each column can be derived from column names or column content, this is not always possible and straightforward. For those reasons, tabular datasets were considered the last "unconquered castle" for DL (Kadra et al., 2021).

The heterogeneous format of tabular data leads to challenges, when one applies DL approaches. It was claimed that traditional ML methods like Gradient-Boosted Decision Trees (GBDT) (Chen & Guestrin, 2016; Dorogush et al., 2018), perform better than DL-based approaches (Kadra et al., 2021; Borisov et al., 2024). Some other researches showed that DL-based approaches are comparable, or even better in certain cases, than tree-based approaches (Huang et al., 2020a; Rubachev et al., 2022). The main disadvantage of tree-based systems is that they typically cannot be trained end-to-end and make use of greedy, local

optimization techniques to build trees. Deep tabular learning can benefit from pre-training tasks, which is the workhorse of DL for vision and language tasks (Rubachev et al., 2022).

This work follows the research direction of applying DL to tabular data. However, we do not aim to solve supervised learning tasks, in which all columns of a table are used to predict another feature. In this work we consider a slightly different problem than Missing Value Imputation (MVI), which we call In-Table Prediction (ITP). Given an uncompleted row of a pre-defined table, in which some values of the columns are missing, the task of ITP is to predict the missing values based on the remaining positions of this row. DL is proposed to solve this problem and we evaluate the performances based on synthetic data.

We consider ITP as a related, but different problem than MVI. MVI replaces the 'NaN' values inside a table. In this work, 'NaN' values are called empty values. The goal of MVI is that the imputed table has no 'NaN' any more and can therefore be processed by down-streaming pipelines, e.g. applying machine learning (ML) algorithm, or computing statistics. This is because the down-streaming pipelines are not compatible with 'NaN' values. ITP does not aim to replace 'NaN' values inside a table, but aims to replace user-defined missing positions in a table. Depending on the context, the user-defined missing positions are called "masked positions" or "missing positions" in this work. The ground-truth (GT) values of masked position can be empty or typical values (a numeric for continuous features or a class ID for categorical features). In this way, ITP assumes that the empty value 'NaN' represents a special status of columns, which should also be recovered during the prediction. In short, MVI replaces empty values 'NaN' for down-streaming pipelines, while ITP replaces masked position for data completion.

ITP has the following usage scenarios. First, ITP can be used to complete the input of a user when a new row of the table is created. During the creation time of a new row, the user could receive suggestions on which values should be used. Second, ITP can be used to recover the data to its original status if some columns of the data are missing or were lost, e.g. due to technical issues. ITP can help to recover the correct values of missing data. Third, by manually inserting masked positions to an existing table, ITP can be used to verify the correctness of the masked positions. Different processes in the world are generating tables every day. Assume that a new table is obtained, ITP could be used to check if some positions of interest conform to the reference data.

Contributions of this work:

(1) Define a slightly different ML problem than MVI for tabular data and compare it with MVI. This problem is called In-Table Prediction (ITP) in this work. In contrast to MVI, the goal of ITP is not to fill out 'NaN' values but to predict the correct values on arbitrarily masked positions. Three usage scenarios are identified for ITP, which cannot be addressed by MVI straightforwardly.

(2) Propose a novel neural layer to embed numeric features and integrate it with the existing DNN to solve IVP for tabular data with continuous features. To ensure methodological rigor and the reliability of the results, this study focuses exclusively on numerical datasets; the inclusion of categorical features is beyond the scope of this work. The continuous features may contain numbers as well as empty values. The proposed neural layer converts them into trainable embeddings, which are fed into the major layers of DNN. Following Gorishniy et al. (2021), MLP, Resnet and Transformer are used as major neural layers in this work.

(3) The proposed structures are applied to solve ITP using generated data. Although related work to pre-train a DNN exists (Rubachev et al., 2022), to our knowledge, nobody has solved ITP by applying NNs. This work proposes a novel structure to address this problem and evaluate the performances based on synthetic data. Compared to the usage of real data, hidden relationships of columns can be predefined in synthetic data, which make it direct to interpret the evaluation results of performances. Our aim is to find out: (1) Can the proposed network learn the predefined relationships regrading to the ground-truth numerical values and the positions of empty values? (2) How good is the performance, regarding to different data size, different types of mechanisms of inserting missing values and different embedding lengths.

## 2 Problem Analysis

The dataset can be formulated as a table, denoted as $T$. $T$ has $N$ rows and $K$ columns. Each column represents a continuous feature and can be denoted as a statistical variable $\mathbf{x}_j$, $j = 1, ..., K$. Each row represents an example, which is assumed to be i.i.d. We use upper index $i$ to refer to a row's index or specific instances, and use lower index $j$ to refer to a column's index. $\mathbf{x} = [\mathbf{x}_1, \ldots, \mathbf{x}_K]$ denotes a vector of continuous features. $x_j^i$ refer to the $j$-th feature for the $i$-th instance. $x_j^i$ locates at the $(i, j)$-th position of table $T$. In this work, we do not consider categorical features.

Importantly, we assume that each $x_j^i$ can take one of the following types: a numerical value, an empty value, namely 'NaN', or a masked value, denoted as $\zeta$. A numeric value indicates that the exact quantity of $\mathbf{x}_j$ is known. An empty value indicates that the value of $x_j$ is missing. A masked value $\zeta$ is just a place holder that is used in training and inference. $\zeta$ presents that the value of this position is not known and it can be a numerical value or an empty value, should be predicted by the model.

According to the position of masked values in $T$, we can partition $T$ into two tables, $T_{full} \neq \emptyset$ and $T_{masked} \neq \emptyset$. All rows in $T_{full}$ contain no masked values, but may contain empty values. All rows in $T_{masked}$ contain at least one masked value and may contain empty values. We assume that there are hidden relationships among the columns in $T_{full}$, e.g. the state of a certain column may depend on the states of other columns. This hidden relationship is, however, not known. Our aim is to train a model on the dataset $T_{full}$, which can predict the values of masked positions in $T_{masked}$, such that the hidden relationship can be fulfilled for $T_{masked}$.

Compared to MVI, the above formulation considers empty values as a special state of $x_j$, which may be subject to certain hidden rules. It requires that the empty values can be predicted when applying the model to table $T_{masked}$. In this way, ITP does not aim to replace empty values by numeric values, but aims to recover the masked positions such that $T_{full}$ and the recovered $T_{masked}$ cannot be differentiated.

## 3 Related Work

**Tabular deep learning** A wide range of architectures and methods have been proposed for tabular deep learning (Yoon et al., 2020; Arik & Pfister, 2019; Huang et al., 2020b; Somepalli et al., 2021; Bahri et al., 2022; Wu et al., 2024; Hollmann et al., 2025). Comprehensive review papers can be found in (Borisov et al., 2024; Gorishniy et al., 2021). Most of these works focused on supervised learning tasks, where the goal is to predict a targeted feature given the columns of a table. Transfer learning for tabular data has also been explored (Zhu et al., 2023; Wang & Sun, 2022), where the datasets used for pretraining and fine tuning differ in table structures and schema. Common characteristics of these approaches include the adaptation of deep learning techniques, which are originally developed for vision and language tasks, such as self-supervised pretraining (e.g. Masked-Language Modeling (Devlin et al., 2018)), the usage of embedding layers to encode a table's columns, transformer architectures with attention mechanisms Vaswani et al. (2017), and autoencoder structures (He et al., 2021). Recent work (Hollmann et al., 2025) also focuses on developing foundation models for tabular data, which can be applied to un-seen tables.

**Self-supervised pretraining** Pretraining is a technique where a model is first trained on a large, general dataset to learn fundamental representations and initial weights before being fine-tuned on a smaller, task-specific dataset. Self-supervised pretraining (Devlin et al., 2018; He et al., 2021; Radford & Narasimhan, 2018) leverages unlabeled data that allow the model to learn useful representations without human annotated labels. In computer vision and natural language processing (NLP), pretraining has become a de facto standard and is considered essential for achieving the state-of-the-art performance (Rubachev et al., 2022). An impactful line of pretraining methods is based on Masked-Language Modeling (MLM) (Devlin et al., 2018) and Masked Autoencoder (MAE) (He et al., 2021), in which a fixed-portion of the input are randomly masked, and the model is trained to reconstruct the masked part. Another line of pretraining methods is based on contrastive learning, which creates negative examples to encourage the model to learn discriminative representations.

Self-supervised pretraining for tabular problems is typically performed directly on the down-streaming target datasets (Huang et al., 2020b; Arik & Pfister, 2019; Yin et al., 2020; Rubachev et al., 2022; Ucar et al., 2021; Zhu et al., 2023; Wang & Sun, 2022). A randomly selected subset of columns are usually masked out, the model is trained to reconstruct the masked values, using the masked values as ground truth. The objective is to learn meaning representations that lead to improved performance on downstream tasks, which is usually performed in a supervised way. In many cases, the pretraining dataset and fine-tuning datasets originate from the same domain, namely the tables have the same structure. However, transfer learning techniques for DTL (Zhu et al., 2023; Wang & Sun, 2022) also incorporate tables from different domains during pretraining, enabling the model to generalize across varying table schemas.

**Missing value imputation** Missing value imputation is the process of replacing missing entries in a dataset with substituted values, aiming to create a complete dataset suitable for data analysis (Ren et al., 2023; Jäger et al., 2021; van Buuren, 2018). Traditional imputation methods include kNNimpute (Troyanskaya et al., 2001), missForest (Stekhoven & Bühlmann, 2011), MICE (van Buuren & Groothuis-Oudshoorn, 2011), which extend classical ML techniques to address MVI problem. In recent years, DL-based generative methods, e.g. Generative Adversarial Network (GAN) and Variational Autoencoder (VAE), have also been employed for MVI (Zhang et al., 2018; Camino et al., 2019; Li et al., 2019; Qiu et al., 2020; Nazabal et al., 2020), offering more flexible and powerful approaches to handle complex missing data patterns.

**Compared to this work** This work builds directly on Gorishniy et al. (2021), which compared the performance of three DL models (MLP, Resnet, transformer) with traditional methods. We adopt the same model types and reuse the modeling structure of FT-Transformer (Gorishniy et al., 2021), which consists of embedding and attention layers. However, compared to Gorishniy et al. (2021), we propose a novel neural layer designed to embed missing values and masks for continuous variables (see Eq. (1)). We solve the ITP based on the technique of MLM-based pretraining, while Gorishniy et al. (2021) does not have a pretraining procedure.

Compared to the previously cited literature for TDL, we do not aim to solve supervised learning tasks, where the objective is typically to predict the value or class of a predefined target feature. We introduce ITP, in which the target features are arbitrary subsets of a table's columns and are not fixed in advance. In our approach both, the masking and training procedure are learned from the general pretraining step of TDL. However, we do not fix the masking ratio and we consider empty values of continuous features.

Compared to MVI methods cited above, ITP does not aim to replace missing values by specific numbers or class labels. Instead, it seeks to recover the original content of masked positions, which can be a number, a class, or empty value. When a table contains only discrete features, ITP can be solved by MVI by treating missing values as a separate class. However, when the table contains continuous features and empty values, ITP can not be solved by standard MVI approaches.

# 4 Our approach

In this section we describe the main architectures and the training strategy utilized in this work.

## 4.1 Model structure

Three types of networks, namely MLP, Resnet and Transformer are adopted as basic building blocks, see Fig. 1. We aim to reuse the well-established building blocks as much as possible and only add task-specific layers before or after these basic building blocks. The basic building blocks of MLP, Resnet are kept the same as the ones used in Gorishniy et al. (2021), which contain linear layers with Relu activation and dropout layers. The building blocks of the transformer-based architecture comes directly from Vaswani et al. (2017), which contains only the encoder part. Because the sequence of table columns should not play a role in predictions, no positional encoding is applied in this work.

The overall structure contains a masking layer, an embedding layer and a basic building block with needed reshape and linear layers. A masking layer is applied first to the input vector $x_1, ..., x_K \in \mathbb{R} \cup \{\text{NaN}\}$ by

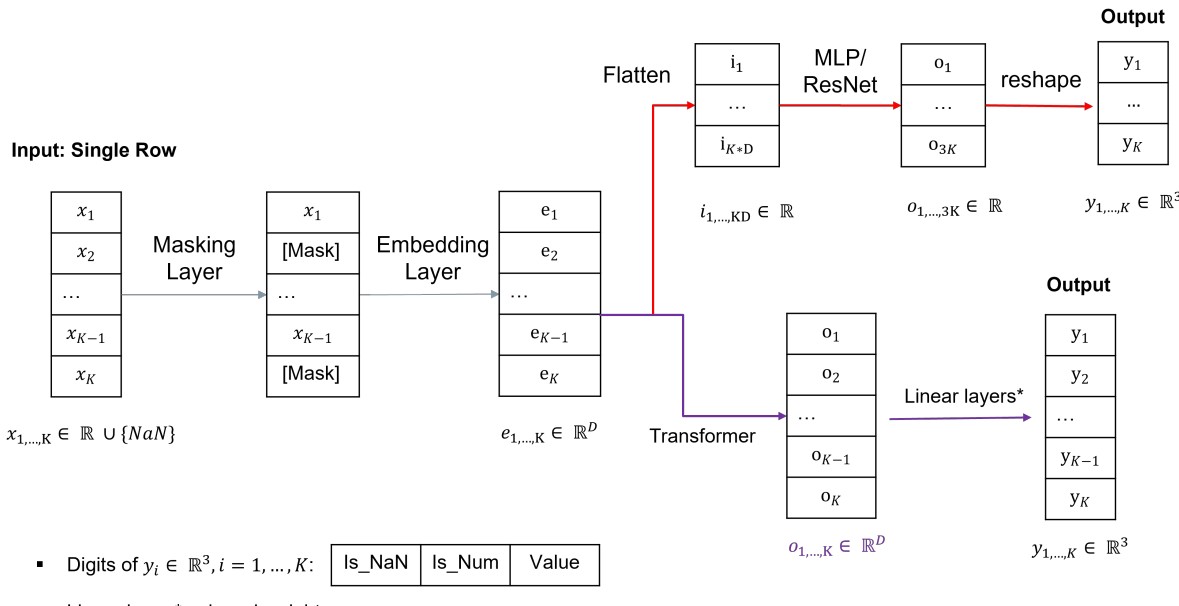

Figure 1: Overall structure of the proposed method. The red route refers to the case, if MLP or Resnet is applied. The blue route refer to the case, if a transformer is applied. Both routes are based on the outputs of two preprocessing steps, namely the masking layer and the embedding layer. For better illustration, we show the case how a single row of table is processed along the entire network with denoted dimensions. During training and testing time, multiple rows can be processed in parallel.

inserting masks. After that, the embedding layer converts numerical values, empty values and masks into $D$-dimensional vectors of trainable variables, denoted as $e_1, ..., e_K \in \mathbb{R}^D$.

If MLP or Resnet is applied, a flatten layer is added to convert all embeddings into a 1-dimensional vector $i = (i_1, ...i_{KD}) \in \mathbb{R}^{3K}$ of length $KD$. $i$ is then consumed by a standard building block directly and outputs $o \in \mathbb{R}^{3K}$ of length $3K$. A final reshaping layer is needed to obtain the final outputs $y_1, ..., y_K \in \mathbb{R}^3$. The first two digits of $y_i$, $i = 1, ..., K$, represent the confidence that the $i$-th column is an empty value. The last digit of $y_i$ is the regressed value, which is valid only if this column is predicted as a number.

If the transformer structure is selected, the input row is processed by a masking and a embedding layer in the same way as above. After that the obtained $K$ embeddings $e_i$ will be processed by the encoder of a transformer without positional encoding. The outputs of the encoder are again a sequence of $K$ vectors denoted as $o_1, ..., o_k$. Finally, the same linear layer is added for each output $o_i$, which outputs vector $y_i \in \mathbb{R}^3$. The meaning of each digit of $y_i$ is the same as before.

The proposed embedding layer is formulated in Eq. (1). For numerical values of column $i$, namely $x_i \in \mathbb{R}$, the embedding strategy follows the method proposed in Gorishniy et al. (2021), where the absolute scalar value $x_i$ is multiplied with the embedding vector $e_{i,1} \in \mathbb{R}^D$. Compared to their work, however, we introduce two additional learnable embedding vectors, $e_{i,2}$ and $e_{i,3}$, for each column $i$ in the embedding layer. $e_{i,2}$ and $e_{i,3}$ represent the embeddings for empty values and masks, respectively. The introduction of these two embeddings is to enable the model to be able to respond to the status of empty values and masks in each column. In sum, the proposed embedding layer can be formulated as

$$f_{embedding}(x_i) = \begin{cases} x_i e_{i,1}, & \text{if } x_i \in \mathbb{R} \\ e_{i,2}, & \text{if } x_i = NaN \\ e_{i,3}, & \text{if } x_i = \zeta \end{cases} \tag{1}$$

where $x_i \in \mathbb{R} \cup \{NaN, \zeta\}$ presents the value of column $i$ after the masking layer. $\zeta$ denotes the mask. $e_{i,1}$, $e_{i,2}$ and $e_{i,3}$ are learnable weights.

## 4.2 Training strategy

The proposed networks generate a sequence of values, which has the same length as the input row, see Fig. 1. During the training, we ask the model to minimize a loss function which corresponds only to the masked positions. During prediction time, the masking layer can be used to fill the unknown positions by masks $\zeta$ and then send the completed row to the model.

In this work, we follow the original idea of BERT Devlin et al. (2018) and minimize the following objective function:

$$L(x,y) = \sum_{\substack{\forall \text{masked } i, \\ i \in \{1,...,K\}}} ce(y_i, x_i) + \mathbf{1}_{x_i \neq \text{NaN}} \cdot (y_{i,2} - x_i)^2 \tag{2}$$

where

$$ce(y_i, x_i) = \begin{cases} -\log y_{i,0}, & \text{if } x_i = \text{NaN} \\ -\log y_{i,1}, & \text{otherwise} \end{cases} \tag{3}$$

$x = (x_1, ..., x_K)^T \in (\mathbb{R} \cup \{\text{NaN}\})^K$ represent a single row of the original table, which contains $K$ values. For $j = 1, ..., K$, $y_j = (y_{j,0}, y_{j,1}, y_{j,2})^T$ represent the model's output for column $j$, refer to Fig. 1. $y_{j,0} \in (0,1)$ and $y_{j,1} \in (0,1)$ represent the predicted possibilities, that position $j$ contains the empty value or not, respectively. $y_{j,2}$ represents the regressed real value. As can be see in Eq. (2), only the masked positions contribute to the loss function and only if a masked position takes numerical value, the term for the regression is activated.

Denote $m_{mask}$ as the number of elements which can be replaced by masks in a single row. Unlike BERT (Devlin et al., 2018) and the pretraining tasks used in DTL (Gorishniy et al., 2021), where a fixed portion of mask is introduced on each example, in this work we allow that all $K - 1$ columns can be masked out during the training process, namely $m_{mask} \leq K - 1$. This setting is motivated by the fact during prediction time it is possible that an user would like to know the rest values given a single inputted variable.

Given limited computational resources, we have fixed all network parameters, including the number of layers, the number neurons in each layer. Hyperparameter tuning of selected model structures is not done in this work. We also fixed the learning rate and applied an early stop strategy to avoid overfitting. If the loss function of the validation set does not decrease for a given number of epochs, the training process is stopped and the best model regarding to the validation set will be saved and evaluated on the test data.

# 5 Design of the experiment

Having introduced the model structure and the training strategy, we now present computational experiments to evaluate the proposed method. To this end, we generate synthetic tabular data, which we then use both for training and for performance evaluation. Unlike real data, synthetic data has the advantage that the underlying relationships among a table's columns can be fully specified and are therefore known a priori. This facilitates a clearer interpretation of predictive performance and enables a more precise assessment of whether the model successfully captures the hidden dependencies among the columns.

## 5.1 Synthetic Data Generation

Synthetic data are generated for training the models and evaluating their performances using the following two steps. In step 1, a complete table without empty values is generated. In step 2, empty values are inserted into the above table.

**Step 1: Generating complete tables** In this step a complete table without empty values are synthetically generated. Having assumed that all columns are subjected to certain hidden relationships, to generate the table we need to first formulate this relationship explicitly.

As we know, shallow NNs are universal approximators (Gühring et al., 2022). They can approximate any continuous multi-variable functions with an arbitrary precision, given that the number of nodes is not limited (Funahashi, 1989; Guliyev & Ismailov, 2018). Although expressivity of NNs has been established in the literature, mathematicians are still analyzing their empirical performance. It is because the existence of a NN that can approximate a target continuous function does not necessarily imply that such a network can be identified through training. There are practical constraints, regarding to data amount, noise, training process, which may affect the performance. Trained NNs can perform very badly on functions for which there are strong expressivity results, such as smooth functions in high dimensions and piecewise smooth functions (Adcock & Dexter, 2021). Research have been done on deducing bounds on how many weights and neurons are necessary for a neural network such that the expressivity can be guaranteed. However, their bounds suffer from the curse of dimensionality and can be week in practice (Gühring et al., 2022).

Following Adcock & Dexter (2021); Gühring et al. (2022), we use the following types of functions to generate tables: (1) a simple linear function, (2) an exponential function of multiple variables. Dataset A is generated by the following linear function

$$x_2 = x_0 + x_1 + \tau d, \tag{4}$$

where $x_0, x_1 \in [-\pi/2, \pi/2]$. $d$ denotes the added noise, which subjects to a standard distribution, i.e. $d \sim \mathcal{N}(0, 1)$. $\tau \in \mathbb{R}$ is a hyper-parameter, which controls the noise level.

Dataset B is generated by using the following exponential function (see (Adcock & Dexter, 2021))

$$x_2 = e^{-\frac{1}{16}(cos(x_0)+cos(x_1))} + \tau d, \tag{5}$$

where $x_0, x_1 \in [0, \pi]$ and $x_2 \in \mathbb{R}$. Using Eq. (5), the generated table contains three continuous columns $x_0$, $x_1$, $x_2$.

**Step 2: Insert empty values** When inserting empty values to a generated completed table, one has to decide the positions of empty values. Following van Buuren (2018), we apply two specific mechanisms of inserting empty values, which belong to the general classes of Missing At Random (MAR) and Missing Not At Random (MNAR). Missing Completely At Random (MCAR) is not evaluated in this work, because there is no hidden relationship for the NN to learn.

Denote $R_j$, $j = 1, \ldots, K$ as a binary statistical variable, representing if the $j$-th column of $\mathbf{x}$ takes an empty value. $P(R_j = 1)$ represents the possibility that $\mathbf{x}_j$ takes empty values, while $P(R_j = 0)$ represents the possibility that $\mathbf{y}_j$ takes numerical values.

Under MAR, we assume that

$$P(R_j = 1) = \mathbf{1}_{K_j}(..., x_l, ...|_{l \neq j}), \forall j = 1, ..., K, \tag{6}$$

where $K_j = \{ (..., x_l, ...)|_{l \neq j} \in \mathbb{R}^{K-1} | \sum_{i \neq j} x_l \geq l \}$ and $l \in \mathbb{R}$ is constant. Eq. (6) means that $\mathbf{x}_j$ takes the empty value, if the summation of the rest features is not less than $l$. $l = 0.5$ is selected and fixed in this work so that a reasonable portion of table elements are replaced by empty values.

Under MNAR, we assume that

$$P(R_j = 1) = \mathbf{1}_{K_j}(x_j), \forall j = 1, ..., K, \tag{7}$$

where $K_j = \{x_j \in \mathbb{R} | x_j \geq 0\}$. Eq. (7) means that $\mathbf{x}_j$ takes the empty value, if its value is no less than 0.

## 5.2 Masking the training, validation and test data

Having introduced the way to generate tabular data with inserted empty values, we now use them to generate training, validation and test data by inserting masks to the generated tables.

For training and validation purpose, two tables are generated, in which a random number of masks are inserted. This number takes the value of 0, 1, ..., $n_{max}$, where $1 \leq n_{max} \leq K - 1$ is a hyperparameter. For example, if we choose $n_{max} = 1$, a maximum of 1 feature can be masked in each row. We call this masking strategy "masking by maximal number $n_{max}$ (MBMN)". The obtained training data is used for updating the network's parameters and the obtained validation data is used for early stopping of the training process.

For the test data, we do not apply the same masking strategy as above. This is because each row of the test table may contain zero, multiple, or all empty values. For the column relationship defined in Eq. (5) and (4), if un-masked positions contain empty values, there is no chance for the model to predict the correct values of the masked positions.

In this work, we apply a different masking strategy to create test data, in order to investigate the question, whether the model learns the column relationship. The idea is that, after the masking process, the values of the masked positions can be fully determined based on the unmasked positions. We call this masking strategy "masking for deterministic relationship (MFDR)". More exactly, for Eq. (5), rows with multiple empty values will be firstly deleted and the remaining rows contain at most one empty value. After that, for each remaining row, if it has a single empty value, this empty value is masked. If it does not have empty values, a single mask is inserted randomly. In this way, the correct values of masked positions can be completely derived from the remaining positions. For Eq. (4), rows which have multiple empty values and rows which have empty values on columns $x_0$, $x_1$ will be firstly deleted. Then column $x_2$ of the remaining table is replaced by masks. Note that in this case we do not insert masks on $x_0$ and $x_1$, because Eq. (4) is not invertible for $x_0$ and $x_1$.

## 5.3 Evaluation metrics

Since continuous features can have different scales, we use Normalized Root Mean Squared Error (NRMSE) (Liew et al., 2010) to evaluate the prediction's performance. Denote $\Theta$ as the positional index of empty values and denote $\Phi$ as the positional index of introduced masks. $\Phi/\Theta$ denotes the index set of masked positions, whose values are not empty.

For any selected column $j^*$,

$$\text{NRMSE}_{j^*} = \sqrt{\frac{\sum_{(i,j^*)\in\Phi/\Theta}(y^i_{j^*} - x^i_{j^*})^2}{\sum_{(i,j^*)\in\Phi/\Theta}(x^i_{j^*})^2}}, \tag{8}$$

where $x^i_j$ and $y^i_j$ denote the ground-truth and predicted values, respectively.

For a entire table,

$$\text{NRMSE} = \sqrt{\frac{\sum_{(i,j)\in\Phi/\Theta}(y^i_j - x^i_j)^2}{\sum_{(i,j)\in\Phi/\Theta}(x^i_j)^2}}, \tag{9}$$

can be used to evaluate the overall performance for all columns.

To evaluate the performance of predicting the positions of empty values, denote $\tilde{\Theta}$ as an index set, which contains the positions of empty values in predictions. $\mathbf{S}_0 = \Theta \cap \tilde{\Theta} \cap \Phi$ refers to the masked positions, which have ground-truth empty values and are predicted as empty values. $\mathbf{S}_1 = \overline{\Theta} \cap \overline{\tilde{\Theta}} \cap \Phi$ refers to the masked positions, which have GT numerical values and are predicted as numerical values. We use accuracy to evaluate the classification performance on empty values.

$$\text{accuracy} = \frac{|\mathbf{S}_0 \cup \mathbf{S}_1|}{|\Phi|} \tag{10}$$

## 6 Results of the experiment

To evaluate the performance of the proposed models, we conduct multiple training experiments with different hyperparameters and their possible values shown in Table 6 of Appendix B. A rigorous grid search strategy is applied, so that each possible combination of the hyperparameter values are tested out. The varied hyperparameters include the applied dataset, the insertion strategy, the model types, the number of inserted masks, the noise level, the training size and the length of the embeddings vectors. The selection of these hyperparameters is guided by the central objective of this study, such as to determine which types of relationships can be learned, which model architecture is most suitable, and how much training data is required.

Table 1: NRMSE and accuracy evaluations on test datasets with different datasets, different maximal numbers of inserted masks ($n_{max} = 1, ..., k - 1$, where $k = 3$ in this case), different model types (MLP, Resnet, attention), different strategies of inserting empty values (MNAR, MAR, NoNaN) and different training sizes (100, 1000, 10000, 50000). Noise level $\tau = 0.01$ and the length of embedding ($D = 512$) is fixed.

| NRMSE | | | MNAR | | | | MAR | | | | NoNaN | | | |
|---|---|---|---|---|---|---|---|---|---|---|---|---|---|---|
| | | | 100 | 1000 | 10000 | 50000 | 100 | 1000 | 10000 | 50000 | 100 | 1000 | 10000 | 50000 |
| x2=x0+x1 | mask<=2 | mlp | 0,839 | 0,203 | 0,139 | 0,124 | 0,756 | 0,231 | 0,120 | 0,112 | 1,004 | 1,001 | 0,183 | 0,113 |
| | | resnet | 0,838 | 0,828 | 0,097 | 0,111 | 0,753 | 0,746 | 0,675 | 0,098 | 1,001 | 0,909 | 0,302 | 0,148 |
| | | attention | 0,565 | 0,214 | 0,718 | 0,138 | 0,515 | 0,237 | 0,106 | 0,073 | 0,554 | 0,196 | 0,094 | 0,069 |
| | mask<=1 | mlp | 0,478 | 0,135 | 0,552 | 0,192 | 0,754 | **0,108** | 0,098 | 0,085 | 1,008 | 1,000 | 0,104 | 0,088 |
| | | resnet | 0,828 | 0,815 | 0,113 | **0,046** | 0,757 | 0,738 | 0,550 | 0,099 | 0,987 | 0,993 | 0,291 | 0,091 |
| | | attention | **0,256** | **0,134** | **0,079** | 0,071 | **0,242** | 0,123 | **0,060** | **0,047** | **0,272** | **0,137** | **0,074** | **0,030** |
| x2=f_exp | mask<=2 | mlp | 0,703 | 0,640 | 0,454 | 0,477 | 0,369 | 0,368 | 0,367 | 0,368 | 0,788 | 0,791 | 0,794 | 0,786 |
| | | resnet | 0,536 | 0,524 | 0,464 | 0,457 | 0,389 | **0,367** | 0,364 | 0,367 | 0,788 | **0,787** | 0,788 | 0,787 |
| | | attention | **0,528** | 0,469 | 0,515 | 0,460 | 0,375 | 0,370 | **0,268** | 0,139 | **0,786** | 0,788 | 0,788 | 0,152 |
| | mask<=1 | mlp | 0,699 | 0,543 | 0,472 | 0,456 | **0,374** | 0,375 | 0,373 | 0,373 | 0,792 | 0,791 | 0,795 | 0,791 |
| | | resnet | 0,537 | 0,527 | 0,351 | 0,197 | 0,378 | 0,378 | 0,370 | 0,378 | 0,788 | **0,787** | 0,788 | 0,787 |
| | | attention | 0,550 | **0,471** | **0,113** | **0,063** | 0,383 | 0,378 | 0,368 | **0,065** | 0,794 | 0,788 | **0,115** | **0,102** |

| Accuracy | | | MNAR | | | | MAR | | | |
|---|---|---|---|---|---|---|---|---|---|---|
| | | | 100 | 1000 | 10000 | 50000 | 100 | 1000 | 10000 | 50000 |
| x2=x0+x1 | mask<=2 | mlp | 0,756 | 0,967 | 0,973 | 0,985 | 0,746 | 0,977 | 0,980 | 0,984 |
| | | resnet | 0,756 | 0,758 | 0,986 | 0,969 | 0,746 | 0,746 | 0,768 | 0,967 |
| | | attention | 0,809 | 0,934 | 0,755 | 0,973 | 0,945 | 0,968 | 0,985 | 0,996 |
| | mask<=1 | mlp | 0,756 | **0,977** | 0,819 | 0,965 | 0,746 | **0,989** | 0,988 | 0,997 |
| | | resnet | 0,756 | 0,758 | 0,984 | **0,989** | 0,746 | 0,746 | 0,980 | **0,998** |
| | | attention | **0,943** | 0,961 | **0,991** | 0,988 | **0,970** | 0,985 | **0,993** | 0,997 |
| x2=f_exp | mask<=2 | mlp | 0,217 | 0,645 | 0,820 | 0,798 | 0,562 | 0,284 | 0,552 | 0,995 |
| | | resnet | 0,645 | 0,645 | 0,811 | 0,816 | 0,562 | 0,788 | 0,325 | 0,985 |
| | | attention | 0,645 | 0,810 | 0,636 | 0,790 | **0,875** | 0,988 | 0,994 | 0,997 |
| | mask<=1 | mlp | 0,217 | 0,649 | 0,769 | 0,795 | 0,326 | 0,324 | 0,890 | 0,993 |
| | | resnet | 0,649 | 0,649 | 0,843 | 0,908 | 0,548 | 0,814 | 0,989 | 0,996 |
| | | attention | **0,649** | **0,817** | **0,978** | 0,968 | 0,548 | **0,995** | 0,995 | **0,999** |

Each training is conducted with a random initialization. Early stopping is implemented for avoiding overfitting (see Section 4.2). Metrics NMSRE in Eq. 9 and accuracy of predicting the positions of empty values in Eq. (10) are reported for the test dataset. Note that due to limited computational capacity, the structures of applied neuron networks are fixed throughout this work, which can be found in Appendix A.

Table 1 presents the experiment results from the grid search approach by changing the hyperparameters, namely the relationships of generating datasets, the maximal numbers of inserted masks, the model types, the strategies of inserting empty values and the training sizes. We conclude that:

1. **Which model structure works best?** For larger datasets (training size $N \geq 10000$), the attention-based structure performs in most cases better than the other two networks (see the bold numbers in Table 1). More specifically, for NRMSE and for the linear dataset, the attention-based network performs better than MLP and Resnet, except for a single case with MNAR strategy and $N = 50000$. For the exponential dataset and based on the NRMSE, the attention-based network outperforms the other two architectures when the training set size exceeds 10,000 samples (for MNAR and NoNaN) and when it is exactly 50,000 samples (for MAR). For accuracy and the exponential dataset, the attention-based network consistently outperforms MLP and Resnet when the training set size exceeds 1000 samples .

2. **How large should be the training size?** The performance increases as the training size $N$ increases. For the linear dataset, starting from a training size of $N = 10000$, the learning process tends to saturate. For

the dataset generated by the exponential function, the saturation effect is not clearly observed for selected values of training size $N$. In this case, we do not rule out the possibility that providing additional training examples could lead to even better performance.

3. **How many masks should be inserted to the training set?** For larger datasets ($N \geq 10000$), the masking strategy "$mask \leq 1$" outperforms "$mask \leq 2$" in most cases. More specifically, for the linear dataset, the masking strategy with $mask \leq 1$ consistently outperforms the strategy with $mask \leq 2$. For the exponential dataset, the strategy with $mask \leq 1$ performs better, except for the case of MAR and $N = 10000$.

4. **Do inserting mechanisms (MNAR, MAR and NoNaN) influence the models' performance?** For the linear dataset with a larger training size ($N = 50000$), there are no significant performance differences among MNAR, MAR and NoNaN. Although the attention-based network performs best, its performance is comparable to that of the other two networks. For the exponential dataset, however, we observe that:

(1) For NoNaN and $N = 50000$, the NRMSE value of Resnet and MLP is significantly higher than that of the attention-based network. This indicates that under NoNaN strategy, MLP and Resnet are unable to learn effectively from the dataset. In contrary, the attention-based network achieves performance comparable to that obtained under MNAR and MAR.

(2) Comparing MNAR and MAR with the same training sizes and $mask \leq 1$, the accuracy of the MNAR strategy is significantly lower than that of MAR. This suggests that MNAR presents a more challenging learning scenario for the proposed networks. We argue that this is probably because under MNAR, the model must first infer the unknown real value and then use it to determine whether the observed value should be missing. This effectively constitutes a two-step, and therefore more complicated, decision process.

Table. 2 presents the experiment results from the grid search approach by changing the length of the embeddings $D$. We can conclude that:

5. **How large should be the length of the embeddings $D$?** For the exponential datasets, varying the embedding dimension $D$ has a negligible impact on the NRMSE of both ResNet and the MLP. For the linear dataset, while the MLP's performance remains largely unaffected by $D$, ResNet exhibits a sensitivity to this parameter in certain cases. In contrast, increasing the embedding length $D$ improves the NRMSE performance of the attention-based network. More specifically, for the linear dataset, the NRMSE performance of the attention-based network begins to saturate at a relatively small $D$, e.g. $D = 128$. For the dataset generated by the exponential function, larger values of $D$ are required for the performance to saturate. For example, in case of MAR and NoNaN, the saturation occurs when $D \geq 256$. In case of MNAR, however, the saturation is not obviously observed for $D \leq 512$.

In summary, in most tested cases, the attention-based structure outperforms MLP and Resnet, when relatively large datasets are available. The inserting strategy with $mask \leq 1$ and large embedding length $D \geq 256$ are also recommended to use. While increasing the embedding length $D$ does not affect effectively the NRMSE performances of MLP and Resnet, it improves the performance of the attention-based network.

## 7 Conclusion

This work investigates the In-Table Prediction (ITP) problem, a novel machine learning problem for tabular data, in which the goal is to predict the values of arbitrarily masked columns in a table based on the remaining known columns. To address ITP, a self-supervised learning approach is proposed, which masks randomly selected columns during training as prediction targets. The study focuses on continuous-feature tables and introduces a novel neural layer that embeds both numeric and missing values as trainable embeddings. These embeddings are integrated with three neural architectures, namely MLP, Resnet, and an attention-based network. Synthetic datasets are generated based on predefined inter-column relationships, with missing values introduced via two mechanisms.

This study compares different model architectures, training set sizes, masking strategies, missingness mechanisms, and embedding dimensions on linear and exponential datasets. Overall, the attention-based architecture achieves the best performance, particularly when larger training sets are available. Performance

Table 2: NRMSE evaluations on test datasets with different datasets, different maximal numbers of inserted masks ($n_{max} = 1, ..., k-1$, where $k = 3$ in this case), different model types (MLP, Resnet, attention), different strategies of inserting empty values (MNAR, MAR, NoNaN) and different lengths of embeddings (32, 128, 256, 512). Noise level $\tau = 0.01$ and the training size $N = 50000$ is fixed in this table.

| NRMSE | | | MNAR | | | | MAR | | | | NoNaN | | | |
|---|---|---|---|---|---|---|---|---|---|---|---|---|---|---|
| | | | 32 | 128 | 256 | 512 | 32 | 128 | 256 | 512 | 32 | 128 | 256 | 512 |
| x2=x0+x1 | mask<=2 | mlp | 0,101 | 0,121 | 0,122 | 0,111 | 0,094 | 0,108 | 0,086 | 0,112 | 0,092 | 0,089 | 0,134 | 0,113 |
| | | resnet | 0,218 | 0,161 | 0,261 | 0,192 | 0,518 | 0,228 | 0,078 | 0,098 | 0,128 | 0,199 | 0,293 | 0,148 |
| | | attention | 0,074 | 0,080 | 0,085 | 0,071 | 0,110 | 0,067 | 0,081 | 0,073 | 0,089 | 0,086 | 0,057 | 0,069 |
| | mask<=1 | mlp | 0,097 | 0,134 | 0,124 | 0,124 | 0,065 | 0,072 | 0,059 | 0,085 | 0,076 | 0,077 | 0,099 | 0,088 |
| | | resnet | 0,059 | 0,121 | 0,112 | 0,138 | 0,201 | 0,178 | 0,140 | 0,099 | 0,064 | 0,131 | 0,174 | 0,091 |
| | | attention | 0,092 | 0,045 | 0,044 | 0,046 | 0,101 | 0,061 | 0,057 | 0,047 | 0,068 | 0,050 | 0,029 | 0,030 |
| x2=f_exp | mask<=2 | mlp | 0,426 | 0,454 | 0,454 | 0,457 | 0,365 | 0,366 | 0,366 | 0,368 | 0,786 | 0,786 | 0,786 | 0,786 |
| | | resnet | 0,467 | 0,469 | 0,455 | 0,456 | 0,368 | 0,369 | 0,367 | 0,367 | 0,787 | 0,787 | 0,787 | 0,787 |
| | | attention | 0,306 | 0,197 | 0,109 | 0,063 | 0,367 | 0,372 | 0,282 | 0,139 | 0,357 | 0,292 | 0,191 | 0,152 |
| | mask<=1 | mlp | 0,471 | 0,475 | 0,483 | 0,477 | 0,369 | 0,369 | 0,369 | 0,373 | 0,794 | 0,794 | 0,788 | 0,791 |
| | | resnet | 0,468 | 0,465 | 0,467 | 0,460 | 0,373 | 0,372 | 0,372 | 0,378 | 0,507 | 0,604 | 0,773 | 0,787 |
| | | attention | 0,395 | 0,347 | 0,331 | 0,197 | 0,375 | 0,148 | 0,064 | 0,065 | 0,526 | 0,204 | 0,098 | 0,102 |

generally improves as the amount of training data increases. For the linear dataset, learning tends to reach a plateau once the dataset becomes sufficiently large, whereas for the exponential dataset no clear saturation is observed, suggesting that additional data may further improve performance.

For the conducted experiments, using a single mask during training is preferable in most cases, especially for larger datasets. The choice of missingness mechanism has little impact on the linear dataset when sufficient data are available, but it plays a more important role for the exponential dataset. In particular, the MNAR mechanism leads to lower accuracy than MAR, indicating a more challenging learning scenario. The embedding dimension has limited influence on MLP and ResNet, but increasing it consistently improves the performance of the attention-based model. Larger embedding sizes are especially beneficial for the exponential dataset.

## A  Applied network structure and training parameters

In this section, the used network structures are documented. The applied network structure for the MLP is shown in Table A. The applied network structure for Resnet is shown in Table A and the applied network structure for the attention-based network is shown in Table A. In this work, the network structures are fixed as defined in the table.

In addition, we fix the following training parameters for our experiments:

- Learning rate = 0.0001

- patient epoches = 20 (early stopping criterion)

Table 3: Applied network structure for the MLP

| Layer | Parameter | Repeated times |
|---|---|---|
| Embedding | $D$ subject to grid search | 1 × |
| Linear | output=128, bias=True | |
| Relu | - | 16 × |
| Dropout | ratio=0.1 | |
| Linear | output=9 | 1 × |

Table 4: Applied network structure for Resnet

| Layer | Parameter | Repeated times |
|---|---|---|
| Embedding | $D$ subject to grid search | 1 × |
| Batch normalization | output=128, momentum=0.1, affine=True | |
| Linear | output=128, bias=True | |
| Relu | - | |
| Dropout | ratio=0.1 | 8 × |
| Linear | output=128, bias=True | |
| Dropout | ratio=0.1 | |
| Batch normalization | output=128, momentum=0.1, affine=True | 1 × |
| Relu | - | 1 × |
| Linear | output=9 | 1 × |

Table 5: Applied network structure for the attention-based network

| Layer | Parameter | Repeated times |
|---|---|---|
| Embedding | $D$ subject to grid search | 1 × |
| Multihead Attention | heads=4, output=32, bias=True | |
| Linear | output=2048, bias=True | |
| Dropout | ratio=0.1 | |
| Linear | output=32, bias=True | 2 × |
| Layer normalization | output=32, elementwise affine=True | |
| Dropout | ratio=0.1 | |
| Linear | output=9 | 1 × |

- batch size = 256

- optimizer = 'adam'

# B  Selected parameters for performance evaluation and comparison

To evaluate the performance of the applied networks and make comparisons (see Section 6), we vary the following parameters: the applied dataset, the insertion strategy, the applied model type, the maximum number of inserted masks $m_{mask}$, the training size and the length of the embeddings $D$. Table 6 lists all selected parameters and their possible values.

A grid search strategy is employed, meaning that training and evaluation are conducted for each possible combination of the candidate values shown in Table 6. The comparison results and discussions are presented in Section 6.

Table 6: Parameters and their possible values for performance evaluation and comparison

| Parameter | Possible Values |
| --- | --- |
| dataset | Eq. (5) or Eq. (4) |
| inserting strategy | MAR, MNAR, NoNaN |
| model type | MLP, Resnet, attention |
| $m_{mask}$ | 1, 2 |
| training size $N$ | 100, 1000, 10000, 50000 |
| $D$ | 32, 128, 256, 512 |

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
