# OpenReview forum: "Solving In-Table Prediction Problems by Deep Neural Networks with Performance Evaluation Using Synthetic Data"
_TMLR — Rejected by TMLR_

### Review · Reviewer_5GYh · 2026-05-11

**Summary Of Contributions:**

The paper compares how three deep learning models (MLP, ResNet, FT-Transformer) perform on a very specific task: given two parametric families of tabular data with three columns containing numeric or NaN values, mask some of those values and predict them back. The authors call this procedure "in-tabular prediction."

The paper has a very limited scope, many issues in its presentation, and makes too many unsupported claims. Everything that is claimed to be novel is, in fact, standard practice in the domain. The scope of the experiments is extremely limited: in the experimental setup of the synthetic training data, in the lack of any valid testing pipeline (test data and train data are essentially identical), and in the choice of models tested.

The most important issue is that the paper is built on top of outdated research and ignores recent developments in the field. Only one recent paper appears to be cited (TabPFN), but it is not considered further. The paper is built on top of the FT-Transformer codebase, which is now 5 years old, in a field where even 1-year-old papers are sometimes already outdated.

**Audience:**

No

**Audience Explanation:**

There is no serious innovation in the paper. The experiments do not add anything to what is already known. The presentation is poor, and recent literature is not cited.

**Claims And Evidence:**

No

**Claims Explanation:**

The authors claim that their "in-tabular prediction" task of masking some values and predicting them back is the innovation of the paper, and identify three usage scenarios that could be summarized as:
1) Autocomplete (propose new values while a user types values in a new row)
2) Missing value imputation (fill in missing data lost due to whatever reason; they first acknowledge this as an existing technique, then essentially describe it as their new application)
3) Data verification / outlier detection (check likelihood of provided values).
However, these are already standard practice, well predating the tabular deep learning era, and cannot be considered in any way new contributions.

The second stated contribution is a novel layer to embed real values with possibly missing / NaN entries, which amounts to a linear 1-dimensional embedding of real values and a learnable flag for NaN entries, plus a separate one for [MASK] tokens. This is, again, standard practice in the field, used in so many papers that it is generally not even mentioned explicitly. For example, the SAINT paper (NeurIPS TRL workshop 2022) and ConTextTab (NeurIPS 2025) both use this approach, among many others.

Finally, and perhaps most importantly, the experimental setup is far too limited to support any broad conclusion. Some of the issues with the experiments:
- The data generation pipeline is extremely narrow and arbitrary. Data always has 3 columns: two uniform i.i.d. variables and a third that is either linear or a specific functional form (plus noise). Some values are then replaced by NaN, but always using linear decision functions.
- Test data is generated with essentially the same pipeline as train data. No serious conclusion can be drawn from this.
- The choice of tested models is extremely limited: one MLP, one ResNet, one transformer, with arbitrarily chosen (and neither explained nor defended) hyperparameters. One cannot judge whether the results are due to the choice of models, the hyperparameters, or something else. The authors cite limited budget, but in that case they should have started from reasonable model sizes, not models with 16+ layers (which are far from clearly beneficial for tabular data).
- No baseline is presented. The described task (predicting masked values, possibly NaN) is in fact just a two-step pipeline: first classification to predict "NaN" (which corresponds to the arbitrary linear decision functions above), followed by regression for those predicted not to be NaN. Given the nature of the tasks, a logistic regression + linear regression approach would most likely outperform all the proposed methods in all but one metric (value prediction with the non-linear data generation pipeline). Any modern tabular foundation model (e.g., TabPFN, TabICL) would almost certainly win in all four tasks, as would classical tree-based ensemble methods.

**Requested Changes:**

In my opinion, to make this work publishable in TMLR or any other reputable venue, the paper would need rewriting to an extent that it would become a different work altogether. The most viable path forward would be to include many more baselines, extend the data generation significantly to dozens of data-generating priors (with justification for why they might be of interest), and thus transform the paper into an evaluation paper for existing methods—checking whether some have particular weaknesses, strengths, etc. This is extensive work, but it could provide valuable insights.

In its present state, I do not see a viable path toward publication. I will still list some presentation problems; many of these issues could have been caught with careful proofreading. In 2026, it is not acceptable to submit a paper in this format without at least some automated proofreading.

- The abstract does not explain what the paper is actually doing, deferring with a generic "Three potential usage scenarios are identified" that one must search for in an unmarked paragraph in the introduction.
- The supposedly central topic of the paper, "In-Table Prediction," has an incorrect acronym ("ITB" instead of "ITP") in the abstract itself.
- The sentence "The building blocks of the transformer-based architecture comes directly from Vaswani et al. (2017), which contains only the encoder part" is clearly incorrect. Perhaps the authors meant that unlike the original transformer, they only use the encoder?
- The depicted path for the ResNet model in Figure 1, though very high-level, is wrong. A ResNet must keep the column and hidden dimensions separate (similar to the transformer path); otherwise it cannot produce an output tensor of the required shape.
- The authors use "digits" where they presumably mean "dimensions."
- The paper claims that BERT-style masking cannot mask all-but-one values, but this is simply wrong. Masking is applied to each element independently with probability p, so with probability $p^{K-1}$ (which is not even particularly low given that K=3) this does happen.
- The beginning of Section 5, "Design of the experiment," consists of a theoretical detour that feels entirely out of place.
- The main result tables appear to be screenshots from MS Excel sheets. In a LaTeX-formatted document, this is not acceptable.
- The 12-page limit is (slightly) exceeded.
- The model outputs 3 dimensions for each token. In fact, 2 suffice, because two of these dimensions are used only for binary classification.
- These two dimensions (out of three) must be followed by a softmax layer, presumably applied implicitly within the cross-entropy loss function, while the remaining value is a regression target learned with L2 loss. However, this is nowhere discussed.

---

### Review · Reviewer_wjuY · 2026-05-23

**Summary Of Contributions:**

**Summary of contributions.**
This paper introduces In-Table Prediction (ITP), a self-supervised learning problem for tabular data where the goal is to predict arbitrarily masked column values from the remaining known columns. The authors propose a novel embedding layer handling numeric, empty (NaN), and masked values as distinct trainable embeddings, and evaluate three architectures (MLP, ResNet, Transformer) on synthetically generated datasets with predefined inter-column relationships. The main finding is that the attention-based architecture outperforms the two others given sufficient training data and large enough embedding dimensions.

**Strengths.**
- The paper makes a clear and useful conceptual distinction between ITP and standard Missing Value Imputation, particularly around the semantic treatment of NaN as a learnable state rather than a value to be replaced. This is a valid and underexplored framing.
- The use of synthetic data with fully controlled ground-truth relationships is a methodologically sound choice for isolating the effect of each hyperparameter, and the grid search across training sizes, masking strategies, and embedding dimensions provides a reasonably systematic empirical analysis.

**Weaknesses.**
- The distinction between ITP and MVI, while conceptually presented, remains insufficiently justified throughout the paper. The three proposed usage scenarios are vague and do not convincingly demonstrate why existing MVI or masked pretraining methods could not address them. The problem framing would benefit from a concrete, grounded example illustrating where ITP strictly outperforms or differs from prior formulations.
- The paper benchmarks only against MLP, ResNet, and the FT-Transformer from Gorishniy et al. (2021), which is a significantly outdated baseline landscape. Notably, Tabular Foundation Models such as TabPFN (Hollmann et al., 2025) and TabICL v2 (Qu et al., 2026)  are directly relevant, yet entirely absent from the experimental comparison. Given that these models are specifically designed for in-context tabular prediction (scenarios closely related to ITP) their omission is a serious gap that undermines the paper's empirical claims.
- The sole architectural novelty, the proposed embedding layer (Eq. 1), amounts to assigning three distinct learned vectors to three token types (numeric, NaN, mask), which is conceptually equivalent to token-type embeddings in BERT (Devlin et al., 2018). This contribution is presented without sufficient discussion of how it differs from or improves upon existing embedding strategies in tabular deep learning.
- Furthermore, the experimental evaluation is limited to synthetically generated tables with only K=3 continuous columns, which severely restricts the generalizability of the conclusions.

**Additional Comments:**

N/A

**Audience:**

No

**Audience Explanation:**

No, because the problem motivation is insufficiently clear and the experimental claims are too narrow to offer actionable insights to the community.

**Claims And Evidence:**

No

**Claims Explanation:**

No, as the evaluation relies solely on synthetic 3-column datasets with no comparison against relevant state-of-the-art methods such as TabPFN or TabICL v2.

**Requested Changes:**

Please see weaknesses above.

---

### Review · Reviewer_tSGD · 2026-05-25

**Summary Of Contributions:**

The paper studies In-Table Prediction (ITP), where arbitrary masked entries in a table are predicted from the remaining entries. The authors distinguish this from missing-value imputation by treating NaN as a valid state to be recovered, rather than a value to be filled. They propose an embedding layer for continuous tabular features that separately represents numerical values, empty values, and mask tokens, and combine it with MLP, ResNet, and Transformer-style architectures. The empirical evaluation is conducted on synthetic linear and exponential three-column datasets with Missing At Random (MAR) / Missing Not At Random (MNAR)-style empty-value mechanisms.

The problem formulation is potentially useful, but the current paper is not convincing as a TMLR submission. The work is evaluated only on very small synthetic settings, lacks real-data experiments and strong baselines, and does not clearly establish novelty over masked tabular modeling, denoising autoencoding, or missing-value modeling. The presentation is also weak: substantial space is spent on generic background, the experimental tables are hard to read, and the paper does not provide enough clean evidence to support the claimed conclusions.

**Audience:**

No

**Audience Explanation:**

The general idea of arbitrary in-table prediction could be interesting to some TMLR readers, especially those working on tabular representation learning or missing-value modeling. However, the actual findings of this paper are limited. The main empirical takeaway is that attention can work better than MLP / ResNet on a small synthetic benchmark when enough training data and embedding dimension are used. This is not a sufficiently novel or useful finding for the broader TMLR audience without real datasets, stronger baselines, ablations, and clearer positioning relative to existing masked tabular learning methods.

**Claims And Evidence:**

No

**Claims Explanation:**

The evidence supports only a much narrower claim: under the authors’ specific synthetic setup, an attention-based model can sometimes outperform MLP and ResNet variants. It does not support the broader claim that the proposed approach effectively solves ITP for tabular data.

My main concern is the experimental setup. The paper uses only two synthetic three-column relationships, one linear and one exponential. This is too limited to demonstrate effectiveness for tabular learning, where real datasets often contain many columns, mixed feature types, noise, irregular missingness, and more complex dependencies. The paper also excludes categorical features, which are central to many tabular datasets.

The baseline comparison is also insufficient. The paper compares only MLP, ResNet, and attention-based variants under the proposed formulation. It does not compare against standard imputation methods, denoising autoencoders, masked tabular pretraining methods, tree-based models, or even simple regression baselines that would be highly relevant for the synthetic linear case. Moreover, the network structures are fixed, and there lacks a study of hyperparameter optimality, making the architecture comparison hard to interpret.

Finally, the presentation of the empirical results is below the expected standard. The main result tables appear as low-resolution, screenshot-like tables and are difficult to inspect. The paper would need much clearer reporting, stronger baselines, and a more realistic evaluation protocol before its claims could be considered well supported.

**Requested Changes:**

The paper should substantially narrow and clarify its claims. The current evidence can at most support conclusions about the two toy synthetic settings tested, not about ITP for tabular data in general.

It is critical to add meaningful baselines, including simple regression baselines for the synthetic functions, standard missing-value imputation methods, denoising autoencoder or masked-reconstruction baselines, and strong tabular-learning models. The paper should also include real-world tabular datasets, preferably with more columns and heterogeneous feature types, or explicitly reposition itself as a preliminary synthetic study.

The authors should provide ablations showing whether the proposed embedding layer is actually responsible for the gains, rather than the choice of architecture or masking protocol. Repeated runs with uncertainty estimates and a fair hyperparameter tuning protocol are also needed.

Finally, the paper needs substantial improvements with respect to its presentation. The result tables should be properly formatted instead of using blurry screenshot-like images. The paper should reduce unnecessary background, cleanly separate main text and appendix material, fix inconsistent terminology, and make the experimental protocol easier to verify.

---

### Author Response · Authors · 2026-06-03
**Response to Reviewers Regarding the Use of Synthetic Data**

We thank all reviewers for their constructive comments and feedback.

A major concern raised by all reviewers is the exclusive use of synthetic data and whether such an evaluation is sufficient for a TMLR publication. We fully acknowledge this concern and agree that demonstrating performance on real-world datasets is important for assessing practical applicability. Nevertheless, we would like to further explain the motivation behind our experimental design.

The primary advantage for using synthetic data is that the underlying relationships among the columns are known by construction. This allows us to directly verify whether the proposed learning framework is capable of learning such relationships and to quantify its performance under controlled conditions. In contrast, for real-world tabular datasets, the underlying dependencies between features are unknown.  Consequently, it can be difficult to determine whether a particular masking configuration yields a uniquely recoverable prediction target, which complicates the interpretation of prediction performance such as MSE.

For example, consider a table with four columns (a), (b), (c) and (d) satisfying a relationship such as (a+b+c+d=0). If two columns (e.g., (a) and (b)) are masked simultaneously, their values cannot be uniquely determined from the remaining columns (c) and (d). In such cases, multiple valid solutions may exist. Consequently, during testing, if we use more than 2 masks and measure the resulted performances, the resulted MSE may not successfully reflect whether the model has successfully captured the underlying structure of the data, since a low error is not necessarily achievable even when the model has learned the correct relationships. More generally, in the real datasets, the recoverability of masked columns from the remaining observed columns is usually unknown. As a result, prediction errors of testing data may reflect both model limitations and intrinsic ambiguity in the prediction task, making performance interpretation more challenging.

Our intention in this work was therefore to first investigate the approach under "controlled" conditions where the ground-truth relationships are explicitly known. This allows us to study the behavior of different neural architectures and masking strategies before addressing more complex and realistic scenarios.

Regarding baselines, we agree that stronger comparisons would further strengthen the study. However, to the best of our knowledge, there is currently no established benchmark dataset or standard evaluation protocol specifically designed for the ITP setting. Since ITP differs from traditional missing-value imputation in both its objective and evaluation procedure, as discussed in the paper, identifying directly comparable baselines is not straightforward.

We acknowledge that future work should include both real-world datasets and a broader set of baseline methods. We view the current study as an initial investigation of the ITP problem under controlled synthetic settings.

---

### Decision · Action_Editor_yF8j · 2026-07-01

**Recommendation:** Reject

**Audience:**

No

**Audience Explanation:**

The conceptual contribution of the paper is interesting, but its novelty is quite limited, as highlighted by several reviewers.

One reviewer also emphasised that the "in-table prediction" is essentially a subclass of more general problems, and I tend to agree. I do not think the case that the problem is significantly different from standard missing data imputation is very convincing at this stage. In general the clarity of the paper should be vastly improved in future iterations of this work.

**Claims And Evidence:**

No

**Claims Explanation:**

There was a consensus among reviewers that the empirical evidence was too weak, in particular because of the simplicity of the synthetic data, and the simplicity of the baselines.